# ITERATIVE DEEP GRAPH LEARNING FOR GRAPH NEURAL NETWORKS

## ABSTRACT

In this paper, we propose an end-to-end graph learning framework, namely **I**terative **D**eep **G**raph **L**earning (IDGL), for jointly learning graph structure and graph embedding simultaneously. We first cast graph structure learning problem as similarity metric learning problem and leverage an adapted graph regularization for controlling smoothness, connectivity and sparsity of the generated graph. We further propose a novel iterative method for searching for hidden graph structure that augments the initial graph structure. Our iterative method dynamically stops when the learned graph structure approaches close enough to the optimal graph. Our extensive experiments demonstrate that the proposed IDGL model can consistently outperform or match state-of-the-art baselines in terms of both downstream task performance and computational time. The proposed approach can cope with both transductive training and inductive training.

## 1 INTRODUCTION

Recent years have seen a significantly growing amount of interest in graph neural networks (GNNs), especially on efforts devoted to developing more effective GNNs for node classification (Li et al., 2016; Kipf & Welling, 2016; Hamilton et al., 2017a), graph classification (Ying et al., 2018b; Ma et al., 2019), and graph generation (Samanta et al., 2018; Li et al., 2018b; You et al., 2018). Encouraged by their huge success, GNNs have widely been used in a variety of domain specific applications such as machine reading comprehension (Chen et al., 2019a), semantic parsing (Xu et al., 2018b), natural language generation (Chen et al., 2019b), and healthcare informatics (Gao et al., 2019).

Despite GNNs' powerful ability of learning expressive node embeddings, unfortunately, GNNs can only be used when graph-structured data is available. Many real-world applications naturally admit network-structured data like social networks or graph-structured data like chemical compounds. However, it is questionable if these intrinsic graph-structures are optimal for the supervised downstream tasks. This is partially because the raw graphs were constructed from the original feature space, which may not reflect the "true" graph topology after feature extraction and transformation. Another potential reason is that real-world graphs are often noisy due to the inevitably error-prone data measurement or collection. More importantly, many applications such as those in natural language processing may only have non-graph structured data or even just the original feature matrix, requiring additional graph construction from the original data matrix to formulate graph data.

Independently, there has been an increasing amount of work studying the dynamic model of interacting systems utilizing implicit interaction models (Sukhbaatar et al., 2016; Hoshen, 2017; Van Steenkiste et al., 2018). There models can be viewed as message passage based graph nerual networks (Gilmer et al., 2017) that pass messages over the fully connected graph through the message passing function (Sukhbaatar et al., 2016) or the use of an attention mechanism (Hoshen, 2017; Van Steenkiste et al., 2018). This has been further extended by Kipf et al. (2018), where they addressed the problem by inferring an explicit interaction structure using a variational graph auto-encoder. However, these methods cannot be directly applicable to joint learning the graph structure and graph representations when the graph is noisy or even not available.

More recently, Franceschi et al. (2019) presented a new approach for jointly learning the graph and the parameters of GNNs, where they learnt a discrete probability distribution on the edges of the graph by approximately solving a bilevel program. Their experimental results have shown promising performance in both cases where the input graph is either corrupted or not available.

However, this approach has severe scalability issue since it needs to learn $N^2$ number of (Bernoulli) random variables to model joint probability distribution on the edges of the graph consisting of $N$ number of vertices. More importantly, it can only be used for transductive setting, which means this method cannot consider new nodes during the testing.

To address these limitations, in this paper, we propose an **I**terative **D**eep **G**raph **L**earning (IDGL) framework for jointly learning the graph structure and the GNN parameters that are optimized towards the prediction task at hand. Our IDGL framework consisting of five parts: 1) a graph learning neural network to generate a graph topology; 2) a graph regularization neural network for controlling the smoothness, connectivity and sparsity of the generated graph; 3) a graph embedding neural network for generating node embeddings; 4) an iterative method to dynamically stop learning when the optimal graph is found; and 5) a prediction neural network for performing a downstream prediction task. In particular, we present a graph learning neural network that casts a graph learning problem as a data-driven similarity metric learning task for constructing a graph. We then adapt techniques for learning graphs from smooth signals (Kalofolias, 2016) to serve as graph regularization. More importantly, we propose a novel iterative method to search for hidden graph structure that augments the initial graph structure toward an optimal graph for the supervised prediction tasks.

We highlight three contributions of our approach as follows:

- We propose an end-to-end graph learning framework for jointly learning graph structure and graph embedding simultaneously. The proposed approach can cope with both transductive training and inductive training.

- We propose to iteratively learn a better graph structure with updated node embeddings, and in the meanwhile, learn better node embeddings with the updated graph structure. In addition, the iterative method dynamically stops when the learned graph structure approaches close enough to the optimal graph based on our proposed stopping criterion.

- We cast graph structure learning problem as similarity metric learning problem and leverage an adapted graph regularization for controlling smoothness, connectivity and sparsity of the generated graph.

- Our extensive experiments demonstrate that our model can consistently outperform or match state-of-the-art baselines.

## 2 AN ITERATIVE DEEP GRAPH LEARNING FRAMEWORK

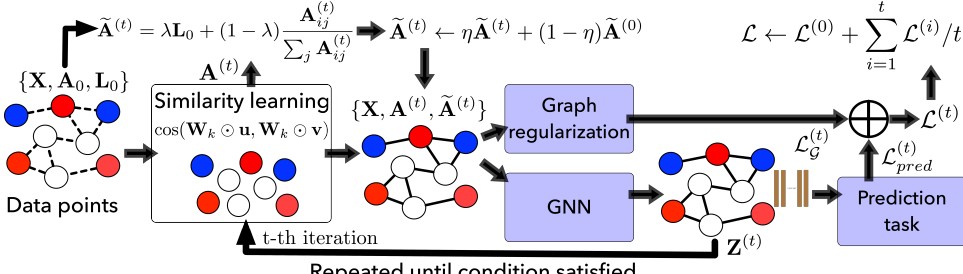

Figure 1: Overview of the proposed model. Dashed lines in the leftmost data points indicate the initial graph topology $\mathbf{A}_0$ either from the ground-truth graph if it exists or otherwise from the graph constructed using the kNN strategy. Best viewed in color.

With this paper we address the challenging problem of automatic graph structure learning for GNNs. We are given a set of $n$ objects $V$ associated with a feature matrix $\mathbf{X} \in \mathbb{R}^{d \times n}$ encoding the feature descriptions of the objects. The goal is to automatically learn the graph structure $\mathcal{G}$, typically in the form of an adjacency matrix $\mathbf{A} \in \mathbb{R}^{n \times n}$, underlying the set of objects, which will be consumed by a GNN-based model for a downstream prediction task.

Unlike most existing methods that construct graphs based on hand-crafted rules or features as a preprocessing step, our proposed **I**terative **D**eep **G**raph **L**earning (IDGL) framework formulates

the problem as an iterative learning problem that jointly learns the graph structure and the GNN parameters iteratively in an end-to-end manner. The overall model architecture is shown in Fig. 1.

## 2.1 GRAPH LEARNING AS SIMILARITY METRIC LEARNING

In traditional graph theory, various methods have been explored to construct a graph from data points. These methods usually apply a metric function to compute the similarity between pairs of nodes during preprocessing, and then consume the constructed graph in a downstream task. Unlike these methods, in this work, we design a learnable metric function for graph structure learning, which will be jointly trained with the prediction model dedicated to a downstream task.

### 2.1.1 SIMILARITY METRIC LEARNING

Common options for metrics include cosine similarity, radial basis function (RBF) kernel and attention mechanisms. A good metric function is supposed to be learnable and expressively powerful. We design a weighted cosine similarity as our metric function, defined as,

$$s_{ij} = \cos(\mathbf{w} \odot \mathbf{v}_i, \mathbf{w} \odot \mathbf{v}_j) \tag{1}$$

where $\odot$ denotes the Hadamard product, and $\mathbf{w}$ is a learnable weight vector which has the same dimension as the input vectors $\mathbf{v}_i$ and $\mathbf{v}_j$, and learns to highlight different dimensions of the vectors.

After preliminary experiments, we have found extending Eq. (1) to a multi-head version to be beneficial, similar to Vaswani et al. (2017); Veličković et al. (2017). Specifically, we use $m$ weight vectors (each vector representing one perspective) to compute $m$ independent similarity matrices using Eq. (1), and take their average as the final similarity $\mathbf{S}$:

$$s_{ij}^k = \cos(\mathbf{w}_k \odot \mathbf{v}_i, \mathbf{w}_k \odot \mathbf{v}_j), \quad s_{ij} = \frac{1}{m}\sum_{k=1}^{m} s_{ij}^k \tag{2}$$

Intuitively, $s_{ij}^k$ computes the cosine similarity between the two input vectors $\mathbf{v}_i$ and $\mathbf{v}_j$, for the $k$-th perspective where each perspective considers one part of the semantics captured in the vectors.

### 2.1.2 GRAPH SPARSIFICATION VIA $\varepsilon$-NEIGHBORHOOD

An adjacency matrix (same for a metric) is supposed to be non-negative while $s_{ij}$ ranges between $[-1, 1]$. In addition, many underlying graph structures are much more sparse than a fully connected graph, which is not only computationally expensive but also makes little sense for most applications. We hence proceed to extract a symmetric sparse adjacency matrix $\mathbf{A}$ from $\mathbf{S}$ by considering only the $\varepsilon$-neighborhood for each node. Specifically, we mask off those elements in $\mathbf{S}$ which are smaller than certain non-negative threshold $\varepsilon$.

$$\mathbf{A}_{ij} = \begin{cases} s_{ij} & s_{ij} > \varepsilon \\ 0 & \text{otherwise} \end{cases} \tag{3}$$

## 2.2 GRAPH REGULARIZATION

In graph signal processing (Shuman et al., 2013), each column of the feature matrix $\mathbf{X}$ can be considered as a graph signal. A widely adopted assumption for graph signals is that values change smoothly across adjacent nodes. Given an undirected graph with symmetric weighted adjacency matrix $A$, the smoothness of a set of graph signals $\mathbf{x}_1, \ldots, \mathbf{x}_n \in \mathbb{R}^d$ is usually measured by the Dirichlet energy (Belkin & Niyogi, 2002),

$$\Omega(\mathbf{A}, \mathbf{X}) = \frac{1}{2}\sum_{i,j} \mathbf{A}_{ij}||\mathbf{x}_i - \mathbf{x}_j||^2 = \text{tr}(\mathbf{X}^T \mathbf{L} \mathbf{X}) \tag{4}$$

where $\text{tr}(\cdot)$ denotes the trace of a matrix, $\mathbf{L} = \mathbf{D} - \mathbf{A}$ is the graph Laplacian, and $\mathbf{D} = \sum_j \mathbf{A}_{ij}$ is the degree matrix. As can be seen, minimizing $\Omega(\mathbf{A}, \mathbf{X})$ forces adjacent nodes to have similar features, thus enforces smoothness of the graph signals on the graph associated to $\mathbf{A}$.

However, solely minimizing the above smoothness loss will result in the trivial solution $\mathbf{A} = 0$. Also, it is desirable to have control of how sparse the resulting graph is. Following Kalofolias

(2016), we impose additional constraints to the learned graph,

$$f(\mathbf{A}) = -\beta \mathbf{1}^T \log(\mathbf{A1}) + \gamma ||\mathbf{A}||_F^2 \tag{5}$$

where $|| \cdot ||_F$ denotes the Frobenius norm of a matrix. As we can see, the first term penalizes the formation of disconnected graphs via the logarithmic barrier, and the second term controls sparsity by penalizing large degrees due to the first term.

In this work, we borrow the above techniques, and apply them as regularization terms to the graph learned by Eqs. (2) and (3). As shown in Eq. (6), the overall graph regularization loss is defined as the sum of the above losses, which is able to control the smoothness, connectivity and sparsity of the resulting graph where $\alpha$, $\beta$ and $\gamma$ are all non-negative hyperparameters.

$$\mathcal{L}_{\mathcal{G}} = \alpha \Omega(\mathbf{A}, \mathbf{X}) + f(\mathbf{A}) \tag{6}$$

## 2.3 ITERATIVE METHOD FOR JOINT GRAPH STRUCTURE AND REPRESENTATION LEARNING

### 2.3.1 JOINT GRAPH STRUCTURE AND REPRESENTATION LEARNING

We expect the graph structure underlying a set of objects to serve two purposes: on the one hand, it should respect the semantic relations among the objects, which is enforced by the metric function (Eq. (2)) and the smoothness loss (Eq. (4)); on the other hand, it should suit the needs of the downstream prediction task.

Compared to previous works which directly optimize the adjacency matrix based on either some graph regularization loss (Kalofolias & Perraudin, 2017), or some task-dependent prediction loss (Franceschi et al., 2019), we propose to learn an optimal similarity metric function as well as the GNN parameters by minimizing a joint loss function combining both the prediction loss defined on the downstream task and the graph regularization loss, namely, $\mathcal{L} = \mathcal{L}_{\text{pred}} + \mathcal{L}_{\mathcal{G}}$.

Note that our graph learning framework is agnostic to various GNNs and prediction tasks. In this paper, we adopt a two-layered GCN (Kipf & Welling, 2016) where the first layer maps the node features to the intermediate embedding space (Eq. (7)), and the second layer further maps the intermediate node embeddings to the output space (Eq. (8)).

$$\mathbf{Z} = \text{ReLU}(\widetilde{\mathbf{A}} \mathbf{X} \mathbf{W}_1) \tag{7}$$

$$\widehat{\mathbf{y}} = \sigma(\widetilde{\mathbf{A}} \mathbf{Z} \mathbf{W}_2) \tag{8}$$

$$\mathcal{L}_{\text{pred}} = \ell(\widehat{\mathbf{y}}, \mathbf{y}) \tag{9}$$

where $\widetilde{\mathbf{A}}$ is the normalized adjacency matrix, $\sigma(\cdot)$ is a task-dependent output function, and $\ell(\cdot)$ is a task-dependent loss function. For instance, for node classification problem, $\sigma(\cdot)$ is a softmax function for predicting a probability distribution over a set of classes, and $\ell(\cdot)$ is a cross-entropy function for computing the prediction loss.

We now discuss how to obtain the normalized adjacency matrix $\widetilde{\mathbf{A}}$. For some problems when an initial graph is available, our preliminary experiments showed that it is harmful to totally discard the initial graph structure. Previous works (Veličković et al., 2017; Jiang et al., 2019) inject the initial graph structure into the graph learning mechanism by performing masked attention, which might limits its graph learning ability. This is because there is no way for their methods to learn weights for those edges that do not exist in the initial graph, but carry useful topological information. With the assumption that the optimal graph structure is potentially a small shift from the initial graph structure, we combine the learned graph structure with the initial graph structure as follows,

$$\widetilde{\mathbf{A}} = \lambda \mathbf{L}_0 + (1 - \lambda) \frac{\mathbf{A}_{ij}}{\sum_j \mathbf{A}_{ij}} \tag{10}$$

where $\mathbf{L}_0$ is the normalized adjacency matrix of the initial graph, defined as, $\mathbf{L}_0 = \mathbf{D}_0^{-1/2} \mathbf{A}_0 \mathbf{D}_0^{-1/2}$, and $\mathbf{D}_0$ is its degree matrix. The adjacency matrix learned by Eqs. (2) and (3) is row normalized such that each row sums to 1. A hyperparameter $\lambda$ is used to balance the trade-off between the learned graph structure and the initial graph structure. If such an initial graph structure is not available, we instead use a kNN graph constructed based on cosine similarity.

---

**Algorithm 1:** IDGL: Iterative Deep Graph Learning Framework

---

**Input:** $\mathbf{X}, \mathbf{y}[, \mathbf{A}_0]$
**Parameters :** $m, \varepsilon, \alpha, \beta, \gamma, \lambda, \delta, T, \eta[, k]$
**Output:** $\Theta, \widetilde{\mathbf{A}}^{(t)}, \widehat{\mathbf{y}}$

1   $[\mathbf{A}_0 \leftarrow \text{kNN}(\mathbf{X}, k)]$        `// Init.` $\mathbf{A}_0$ `to kNN graph if` $\mathbf{A}_0$ `is unavailable`
2   $\mathbf{A}^{(0)}, \widetilde{\mathbf{A}}^{(0)} \leftarrow \{\mathbf{X}, \mathbf{A}_0\}$ using Eqs. (2), (3) and (10)        `// Learn the adj. matrix`
3   $\mathbf{Z}^{(0)} \leftarrow \{\widetilde{\mathbf{A}}^{(0)}, \mathbf{X}\}$ using Eq. (7)        `// Compute node embeddings`
4   $\mathcal{L}_{\text{pred}}^{(0)} \leftarrow \{\widetilde{\mathbf{A}}^{(0)}, \mathbf{Z}^{(0)}, \mathbf{y}\}$ using Eqs. (8) and (9)        `// Compute prediction loss`
5   $\mathcal{L}_{\mathcal{G}}^{(0)} \leftarrow \{\mathbf{A}^{(0)}, \mathbf{X}\}$ using Eqs. (4)–(6)        `// Compute graph regularization loss`
6   $\mathcal{L}^{(0)} \leftarrow \mathcal{L}_{\text{pred}}^{(0)} + \mathcal{L}_{\mathcal{G}}^{(0)}$        `// Compute joint loss`
7   $t \leftarrow 0$
8   **while** $(t == 0 \quad or \quad ||\mathbf{A}^{(t)} - \mathbf{A}^{(t-1)}||_F^2 > \delta ||\mathbf{A}^{(0)}||_F^2) \quad and \quad t < T$ **do**
9      $t \leftarrow t + 1$
10     $\mathbf{A}^{(t)}, \widetilde{\mathbf{A}}^{(t)} \leftarrow \{\mathbf{Z}^{(t-1)}, \mathbf{A}_0\}$ using Eqs. (2), (3) and (10)        `// Refine the adj. matrix`
11     $\bar{\mathbf{A}}^{(t)} \leftarrow \{\widetilde{\mathbf{A}}^{(t)}, \widetilde{\mathbf{A}}^{(0)}\}$ using Eq. (11)
12     $\mathbf{Z}^{(t)} \leftarrow \{\bar{\mathbf{A}}^{(t)}, \mathbf{X}\}$ using Eq. (7)        `// Refine node embeddings`
13     $\widehat{\mathbf{y}} \leftarrow \{\bar{\mathbf{A}}^{(t)}, \mathbf{Z}^{(t)}\}$ using Eq. (8)        `// Compute task output`
14     $\mathcal{L}_{\text{pred}}^{(t)} \leftarrow \{\widehat{\mathbf{y}}, \mathbf{y}\}$ using Eq. (9)
15     $\mathcal{L}_{\mathcal{G}}^{(t)} \leftarrow \{\mathbf{A}^{(t)}, \mathbf{X}\}$ using Eqs. (4)–(6)
16     $\mathcal{L}^{(t)} \leftarrow \mathcal{L}_{\text{pred}}^{(t)} + \mathcal{L}_{\mathcal{G}}^{(t)}$
17 **end**
18   $\mathcal{L} \leftarrow \mathcal{L}^{(0)} + \sum_{i=1}^{t} \mathcal{L}^{(i)}/t$
19 **if** *Training* **then**
20     Back-propagate $\mathcal{L}$ to update model weights $\Theta$
21 **end**

---

### 2.3.2 ITERATIVE METHOD FOR GRAPH LEARNING

Some previous works (Veličković et al., 2017) rely solely on raw node features to learn the graph structure based on some attention mechanism, which we think have some limitations since raw node features might not contain enough information to learn good graph structures. Our preliminary experiments showed that simply applying some attention function upon these raw node features does not help learn meaningful graphs (i.e., attention scores are kind of uniform). Even though we train the model jointly using the task-dependent prediction loss, we are limited by the fact that the similarity metric is computed based on the potentially inadequate raw node features.

To address the above limitation, we propose an Iterative Deep Graph Learning (IDGL) framework. A sketch of the IDGL framework is presented in Algorithm 1. Inputs and operations in squared brackets are optional. Specifically, besides computing the node similarity based on their raw features, we further introduce another learnable similarity metric function ( Eq. (2)) that is rather computed based on the intermediate node embeddings, as demonstrated in Line 10. Compared to the raw node features, these intermediate node embeddings usually reside on a low-dimensional manifold of the raw node feature space, and are optimized towards the downstream prediction task. The aim is that the metric function defined on this node embedding space is able to learn topological information supplementary to the one learned solely based on the raw node features. In order to combine the advantages of both the raw node features and the intermediate node embeddings, we make the final learned graph structure as a linear combination of both of them,

$$\bar{\mathbf{A}}^{(t)} = \eta \widetilde{\mathbf{A}}^{(t)} + (1 - \eta) \widetilde{\mathbf{A}}^{(0)} \tag{11}$$

where $\widetilde{\mathbf{A}}^{(t)}$ and $\widetilde{\mathbf{A}}^{(0)}$ are the two normalized adjacency matrices learned by Eq. (10) at the $t$-th iteration and the initialization step before the iterative loop, respectively.

Furthermore, as we can see from Line 10 to Line 12, the algorithm repeatedly refines the adjacency matrix $\widetilde{\mathbf{A}}^{(t)}$ with the updated node embeddings $\mathbf{Z}^{(t-1)}$, and in the meanwhile, refines the node embeddings $\mathbf{Z}^{(t)}$ with the updated adjacency matrix $\widetilde{\mathbf{A}}^{(t)}$. The iterative procedure dynamically stops

when the learned adjacency matrix converges (with certain threshold $\delta$) or the maximal number of iterations is reached ( Line 8). Compared to using a fixed number of iterations globally, the advantage of applying this dynamical stopping strategy becomes more clear when we are doing mini-batch training since we can adjust when to stop dynamically for each example graph in the mini-batch. At each iteration, a joint loss combining both the task-dependent prediction loss and the graph regularization loss is computed ( Line 16). After all iterations, the overall loss will be back-propagated through all previous iterations to update the model parameters (Line 20).

## 2.4 FORMAL ANALYSIS

### 2.4.1 CONVERGENCE OF THE ITERATIVE LEARNING PROCEDURE

While it is challenging to theoretically prove the convergence of the proposed iterative learning procedure due to the arbitrary complexity of the involved learning model, here we want to conceptually understand why it works in practice. Fig. 2 shows the information flow of the learned adjacency matrix $\mathbf{A}$ and the intermediate node embedding matrix $\mathbf{Z}$ during the iterative procedure. For the sake of simplicity, we omit some other variables such as $\tilde{\mathbf{A}}$. As we can see, at $t$-th iteration, $\mathbf{A}^{(t)}$ is computed based on $\mathbf{Z}^{(t-1)}$ (Line 10), and $\mathbf{Z}^{(t)}$ is computed based on $\tilde{\mathbf{A}}^{(t)}$ (Line 12) which is computed based on $\mathbf{A}^{(t)}$ (Eq. (10)). We further denote the difference between the adjacency matrices at the $t$-th iteration and the previous iteration by $\delta_A^{(t)}$. Similarly, we denote the difference between the node embedding matrices at the $t$-th iteration and the previous iteration by $\delta_Z^{(t)}$.

If we assume that $\delta_Z^{(1)} < \delta_Z^{(0)}$, then we can expect that $\delta_A^{(2)} < \delta_A^{(1)}$ because conceptually more similar node embedding matrix (i.e., smaller $\delta_Z$) is supposed to produce more similar adjacency matrix (i.e., smaller $\delta_A$) given the fact that model parameters keep the same through iterations. Similarly, given that $\delta_A^{(2)} < \delta_A^{(1)}$, we can expect that $\delta_Z^{(2)} < \delta_Z^{(1)}$. Following this chain of reasoning, we can easily extend it to later iterations. In order to see why the assumption $\delta_Z^{(1)} < \delta_Z^{(0)}$ makes sense in practice, we need to recall the fact that $\delta_Z^{(0)}$ measures the difference between $\mathbf{Z}^{(0)}$ and $\mathbf{X}$, which is usually larger than the difference between $\mathbf{Z}^{(1)}$ and $\mathbf{Z}^{(0)}$, namely $\delta_Z^{(1)}$. We will empirically examine the convergence property of the iterative learning procedure in the experimental section.

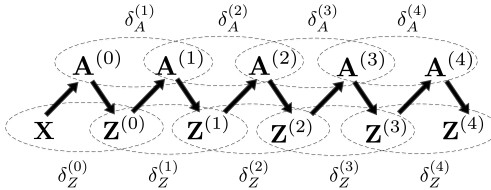

Figure 2: Information flow of the proposed iterative learning procedure.

### 2.4.2 MODEL COMPLEXITY

The cost of learning an adjacency matrix is $\mathcal{O}(n^2 h)$ for $n$ nodes and data in $\mathbb{R}^h$, while computing node embeddings costs $\mathcal{O}(n^2 d + ndh)$, computing task output costs $\mathcal{O}(n^2 h)$, and computing the total loss costs $\mathcal{O}(n^2 d)$. We set the maximal number of iterations to $T$, hence the overall complexity is $\mathcal{O}(Tn(nh + nd + hd))$. If we assume that $d \approx h$ and $n \gg d$, the overall complexity is $\mathcal{O}(Tdn^2)$.

## 3 EXPERIMENTS

In this section, we conducted a series of experiments to answer three main questions. First, when the graph topology is available, can IDGL further improve performance on downstream tasks by learning additional topological information supplementary to the given topology? We also wanted to examine if IDGL is robust to noisy graphs with random edge deletions or additions. Second, can IDGL achieve reasonably good results on semi-supervised learning problems for which a graph is not available. Third, what is the performance of IDGL on inductive learning problems where there are new nodes during the testing?

In addition to answering the above three questions, we also analyzed the proposed model in multiple dimensions. First, we conducted an ablation study to assess the impact of different model components. Second, we empirically examine whether the iterative learning procedure can converge in practice. Third, we explored and compared different stopping strategies that can be used in the iterative learning framework. Finally, we empirically compared IDGL with LDS (Franceschi et al., 2019) in terms of training efficiency. The details on model settings are provided in Appendix C. The implementation of our model will be made publicly available upon the acceptance of this paper.

## 3.1 DATASETS

The benchmarks used in our experiments include two network benchmarks, three data point benchmarks and two text benchmarks. Cora and Citeseer are two commonly used network benchmarks for evaluating graph-based learning algorithms (Sen et al., 2008). The input features are bag of words and the task is node classification. In addition to Cora and Citeseer where the graph topology is available, we evaluate IDGL on three data point benchmarks (i.e., Wine, Breast Cancer (Cancer) and Digits from the UCI machine learning repository (Dua & Graff, 2017)). The task is also node classification. Finally, to demonstrate the effectiveness of IDGL on inductive learning problems, we conduct document classification and regression tasks on the 20Newsgroups data (20News) and the movie review data (MRD) (Pang & Lee, 2004), respectively. In this setting, we regard each document as a graph containing each word as a node. Please refer to Appendix B for data statistics.

## 3.2 SETUP AND BASELINES

For Cora and Citeseer, we follow the experimental setup of previous works (Kipf & Welling, 2016; Veličković et al., 2017; Franceschi et al., 2019). For Wine, Cancer and Digits, we follow the experimental setup of Franceschi et al. (2019). For 20News, we randomly select 30% examples from the training data as the development set. For MRD, we split the data to train/dev/test sets using a 60%/20%/20% split. The reported results are averaged over 5 runs with different random seeds.

Our main baseline in the transductive setting is LDS. Similar to our work, LDS also jointly learns the graph structure and the parameters of GNNs. However, LDS is incapable of handling inductive learning problems since it aims at directly optimizing the discrete probability distribution on the edges of the underlying graph, which makes it unable to handle unseen nodes/graphs in the testing phase. The experimental results of several semi-supervised (e.g., label propagation (LP) (Zhu et al., 2003), manifold regularization (ManiReg) (Belkin et al., 2006), semi-supervised embedding (SemiEmb) (Weston et al., 2012)) and supervised learning (logistic regression (LogReg), support vector machines (Linear and RBF SVM), random forests (RF), and feed-forward neural networks (FFNN)) baselines are reported in the LDS paper. For the sake of completeness, we directly copy their results here. For ease of comparison, we also copy the reported results of LDS even though we rerun the experiments of LDS using the official code released by the authors.

In addition, for Cora and Citeseer, we include GCN (Kipf & Welling, 2016) and GAT (Veličković et al., 2017) as baselines. In order to evaluate the robustness of IDGL to noisy graphs, we also compare IDGL with GCN on graphs with edge deletions or additions. For data point benchmarks where the graph topology is not available, we conceive a kNN-GCN baseline where a kNN affinity graph on the data set is first constructed as a preprocessing step before applying a GCN. For 20News and MRD in the inductive setting, we compare IDGL with a BiLSTM (Hochreiter & Schmidhuber, 1997) baseline and kNN-GCN.

## 3.3 EXPERIMENTAL RESULTS

The results of transductive and inductive experiments are shown in Table 1 and Table 2. First of all, we can see that IDGL outperforms all baseline methods in 6 out of 7 benchmarks, which demonstrates the effectiveness of the proposed method. Besides, by comparing the results of GCN, GAT and IDGL on Cora and Citeseer, and considering the fact that our method is actually based on GCN, we can conclude that our graph learning method can greatly help the node classification task even when the graph topology is given. When the graph topology is not given, we observe that kNN-GCN works well and provides competitive results compared to the supervised baselines that do not leverage graph structures. This indicates the benefits of learning and exploiting underlying graph

Table 1: Test accuracy ($\pm$ standard deviation) in percentage on various classification datasets in the transductive setting. The star symbol means that we run the experiments and report the results.

| Methods | Cora | Citeseer | Wine | Cancer | Digits |
|---|---|---|---|---|---|
| LogReg | 60.8 (0.0) | 62.2 (0.0) | 92.1 (1.3) | 93.3 (0.5) | 85.5 (1.5) |
| Linear SVM | 58.9 (0.0) | 58.3 (0.0) | 93.9 (1.6) | 90.6 (4.5) | 87.1 (1.8) |
| RBF SVM | 59.7 (0.0) | 60.2 (0.0) | 94.1 (2.9) | 91.7 (3.1) | 86.9 (3.2) |
| RF | 58.7 (0.4) | 60.7 (0.7) | 93.7 (1.6) | 92.1 (1.7) | 83.1 (2.6) |
| FFNN | 56.1 (1.6) | 56.7 (1.7) | 89.7 (1.9) | 92.9 (1.2) | 36.3 (10.3) |
| LP | 37.8 (0.2) | 23.2 (6.7) | 89.8 (3.7) | 76.6 (0.5) | 91.9 (3.1) |
| ManiReg | 62.3 (0.9) | 67.7 (1.6) | 90.5 (0.1) | 81.8 (0.1) | 83.9 (0.1) |
| SemiEmb | 63.1 (0.1) | 68.1 (0.1) | 91.9 (0.1) | 89.7 (0.1) | 90.9 (0.1) |
| LDS | 84.1 (0.4) | 75.0 (0.4) | 97.3 (0.4) | 94.4 (1.9) | 92.5 (0.7) |
| GCN | 81.0 (0.2) | 70.9 (0.3) | — | — | — |
| GAT | 82.5 (0.4) | 70.9 (0.4) | — | — | — |
| kNN-GCN | — | — | 95.9 (0.9) | 94.7 (1.2) | 89.5 (1.3) |
| LDS* | 83.9 (0.6) | **74.8 (0.3)** | 96.9 (1.4) | 93.4 (2.4) | 90.8 (2.5) |
| IDGL | **84.5 (0.3)** | 74.1 (0.2) | **97.8 (0.6)** | **95.1 (1.0)** | **93.1 (0.5)** |

Table 2: Test scores ($\pm$ standard deviation) in percentage on classification (accuracy) and regression ($R^2$) datasets in the inductive setting.

| Methods | 20News | MRD |
|---|---|---|
| BiLSTM | 80.0 (0.4) | 53.1 (1.4) |
| kNN-GCN | 81.3 (0.6) | 60.1 (1.5) |
| IDGL | **83.6 (0.4)** | **63.7 (1.8)** |

structures. Compared to kNN-GCN, IDGL consistently achieves much better results on all datasets, which shows the power of jointly learning graph structures and GNN parameters. Compared to LDS, IDGL achieves better performance in 4 out of 5 benchmarks. Unlike LDS which can only handle transductive setting, IDGL can easily handle inductive setting without a modification of the algorithm. This is because IDGL aims at optimizing a metric function instead of the discrete probability distribution on the edges. The good performance on 20News and MRD verifies the capability of IDGL on inductive learning problems.

To evaluate the robustness of IDGL on noisy graphs, we construct graphs with random edge deletions or additions. Specifically, we randomly remove or add 25%, 50% and 75% of the edges in the original graphs. The results on the edge deletion graphs and edge addition graphs are shown in Fig. 3 and Fig. 4, respectively. As we can clearly see, compared to GCN, IDGL achieves better results in all scenarios and is much more robust to noisy graphs. While GCN completely fails in the edge addition scenario, IDGL is still able to perform reasonably well. We conjecture this is because Eq. (10) is formulated in a form of skip-connection, by lowering the value of $\lambda$, we enforce the model to rely less on the initial noisy graph that contains too much additive random noise.

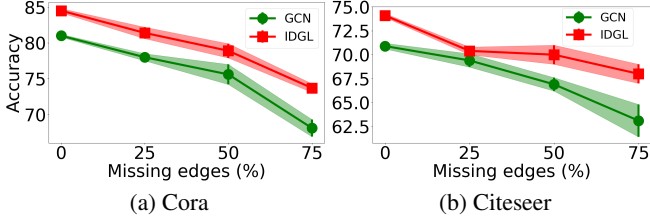

(a) Cora        (b) Citeseer

Figure 3: Test accuracy ($\pm$ standard deviation) in percentage for the edge deletion scenario.

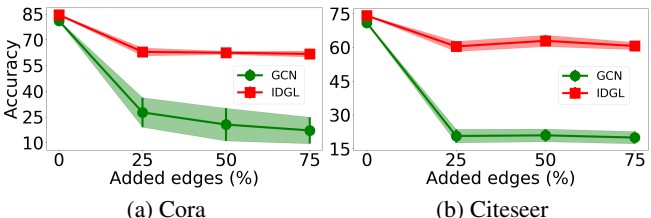

(a) Cora        (b) Citeseer

Figure 4: Test accuracy ($\pm$ standard deviation) in percentage for the edge addition scenario.

## 3.4 ABLATION STUDY

We perform an ablation study to assess the impact of different model components. As shown in Table 3, we can see a significant performance drop consistently on all datasets (e.g., 3.1% on Citeseer) by turning off the iterative learning component, which demonstrates the effectiveness of the proposed iterative learning framework for the graph learning problem. We can also see the benefits of jointly training the model with the graph regularization loss. For instance, when training the model without the graph regularization loss, the performance on Citeseer drops from 74.1% to 71.5%.

Table 3: Ablation study on various classification datasets.

| Methods | Cora | Citeseer | Wine | Cancer | Digits | 20News |
|---|---|---|---|---|---|---|
| IDGL | **84.5 (0.3)** | **74.1 (0.2)** | **97.8 (0.6)** | **95.1 (1.0)** | **93.1 (0.5)** | **83.6 (0.4)** |
| w/o graph reg. | 84.3 (0.4) | 71.5 (0.9) | 97.3 (0.8) | 94.9 (1.0) | 91.5 (0.9) | 83.4 (0.5) |
| w/o IL | 83.5 (0.6) | 71.0 (0.8) | 97.2 (0.8) | 94.7 (0.9) | 92.4 (0.4) | 83.0 (0.4) |

## 3.5 MODEL ANALYSIS

In Fig. 5, we show the evolution of the learned adjacency matrix and accuracy through iterations in the iterative learning procedure in the testing phase. We compute the difference between adjacency matrices at consecutive iterations as $\delta_A^{(t)} = ||\mathbf{A}^{(t)} - \mathbf{A}^{(t-1)}||_F^2 / ||\mathbf{A}^{(t)}||_F^2$ which typically ranges from 0 to 1. As we can see, both the adjacency matrix and accuracy converge quickly through iterations. This empirically verifies the analysis we made on the convergence property of the iterative learning procedure in Section 2.4.1.

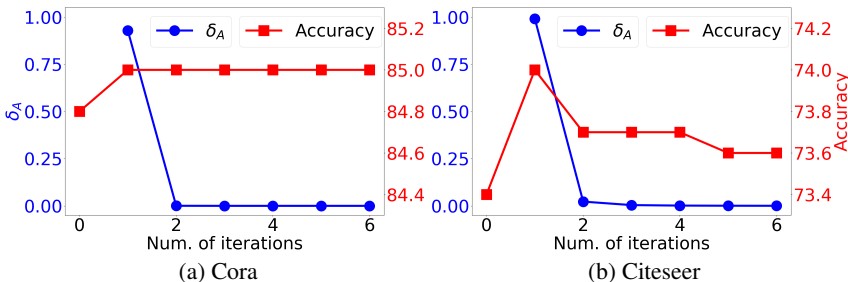

(a) Cora                    (b) Citeseer

Figure 5: Evolution of the learned adjacency matrix and test accuracy (in %) through iterations in the iterative learning procedure.

There are two natural ways of designing the stopping strategy for iterative learning methods. We can either use a fixed number of iterations, or dynamically determine if the learning procedure already converges or not based on some stopping criterion. In Fig. 6, we empirically compare the effectiveness of the above two strategies. We run IDGL on Cora (left) and Citeseer (right) using different stopping strategies with 5 runs, and report the average accuracy. As we can see, dynamically adjusting the number of iterations using the stopping criterion works better in practice.

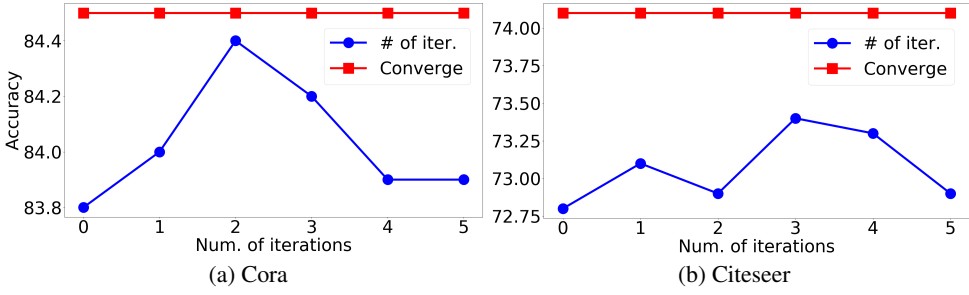

(a) Cora                    (b) Citeseer

Figure 6: Performance comparison (i.e., test accuracy in %) of two different stopping strategies: i) using a fixed number of iterations (blue line), and ii) using a stopping criterion to dynamically determine the convergence (red line).

## 3.6 TIMING

Finally, we compare the training efficiency of IDGL, LDS and other classic GNNs (e.g., GCN and GAT) on various benchmarks. All experiments are conducted on the same machine which has an Intel i7-2700K CPU, an Nvidia Titan Xp GPU and 16GB RAM, and are repeated 5 times with different random seeds. Results are shown in Table 4. As we can see, both IDGL and LDS are slower than GCN and GAT, which is as expected since GCN and GAT do not need to learn graph structures simultaneously. IDGL is consistently faster than LDS, but in general, they are comparable. We also find that the iterative learning part is the most time consuming in IDGL.

Table 4: Mean and standard deviation of training time on various benchmarks (in seconds).

| Benchmarks | Cora | Citeseer | Wine | Cancer | Digits |
|---|---|---|---|---|---|
| GCN | 3 (1) | 5 (1) | — | — | — |
| GAT | 26 (5) | 28 (5) | — | — | — |
| LDS | 390 (82) | 585 (181) | 33 (15) | 25 (6) | 72 (35) |
| IDGL | 237 (21) | 563 (100) | 20 (7) | 21 (11) | 65 (12) |
| IDGL w/o IL | 49 (8) | 61 (15) | 3 (2) | 3 (1) | 2 (1) |

## 4 RELATED WORK

In the field of graph signal processing, researchers have explored various ways of learning graphs from data (Dong et al., 2016; Kalofolias, 2016; Kalofolias & Perraudin, 2017; Egilmez et al., 2017), with certain assumptions (e.g., smoothness) on the graph signals or structural constraints (e.g., connectivity and sparsity) on the underlying graphs. Notably, these works in general do not consider any downstream task that will consume the learned graph structures.

Over the past few years, graph neural networks (GNNs) (Kipf & Welling, 2016; Gilmer et al., 2017; Hamilton et al., 2017b; Li et al., 2015) have drawn increasing attention, and have many successful applications in computer vision (Norcliffe-Brown et al., 2018), natural language processing (Xu et al., 2018a;b;c) and recommender systems (Ying et al., 2018a). How to apply GNNs to applications where the underlying graph structures are unavailable becomes an emergent and challenging problem. However, manually constructing graphs from data heavily relies on domain knowledge and is not very scalable. Very recently, researchers have explored methods to automatically construct a graph of objects (Norcliffe-Brown et al., 2018; Choi et al., 2019; Franceschi et al., 2019; Li et al., 2018a) or words (Liu et al., 2018; Chen et al., 2019a;b) when applying GNNs to non-graph structured data. However, these methods merely optimize the graphs towards the downstream tasks without utilizing the techniques which have proven to be useful in graph signal processing.

More recently, Franceschi et al. (2019) proposed the LDS model for jointly learning the graph and the parameters of GNNs by leveraging the bilevel optimization technique. Unlike LDS, in our model, we optimize a joint loss combining both task-specific prediction loss and graph regularization loss. Moreover, we propose to repeatedly refine the learned adjacency matrix and the node embeddings via an iterative deep graph learning framework. It will be interesting to see if our model can benefit from adopting the bilevel optimization technique, which can motivate future work. However, one big limitation of the LDS work is that it optimizes the discrete probability distribution on the edges of the graph, which makes it unable to handle the inductive setting.

## 5 CONCLUSION

In this paper, we proposed an Iterative Deep Graph Learning (IDGL) framework for jointly learning the graph structure and the GNN parameters that are optimized towards the prediction task. The proposed method is able to iteratively search for hidden graph structures that better help the downstream prediction task. We cast graph structure learning problem as similarity metric learning problem and leverage an adapted graph regularization for controlling smoothness, connectivity and sparsity of the generated graph. Our extensive experiments demonstrate that the proposed IDGL model can consistently outperform or match state-of-the-art baselines in terms of both classification accuracy and computational time. We leave how to design better metric functions as future work.

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

## A  EFFECT OF $\lambda$

Table 5: Test scores ($\pm$ standard deviation) with different values of $\lambda$ on the Cora data.

| Methods / $\lambda$ | 0.9 | 0.8 | 0.7 | 0.6 | 0.5 |
|---|---|---|---|---|---|
| IDGL | 83.6 (0.4) | 84.5 (0.3) | 83.9 (0.3) | 82.4 (0.1) | 80.9 (0.2) |

## B  DATA STATISTICS

Table 6: Data statistics.

| Benchmarks | Train/Dev/Test | Task | Setting |
|---|---|---|---|
| Cora | 140/500/1,000 | node classification | transductive |
| Citeseer | 120/500/1,000 | node classification | transductive |
| Wine | 10/20/158 | node classification | transductive |
| Cancer | 10/20/539 | node classification | transductive |
| Digits | 50/100/1,647 | node classification | transductive |
| 20News | 7,919/3,395/7,532 | graph classification | inductive |
| MRD | 3,003/1,001/1,002 | graph regression | inductive |

## C  MODEL SETTINGS

In all our experiments, we apply a dropout ratio of 0.5 after GCN layers except for the output GCN layer. During the iterative learning procedure, we also apply a dropout ratio of 0.5 after the intermediate GCN layer, except for Citeseer (no dropout) and Digits (0.3 dropout). For experiments on text benchmarks, we keep and fix the 300-dim GloVe vectors for words that appear more than 10 times in the dataset. For long documents, for the sake of efficiency, we cut the text length to maximal 1,000 words. We apply a dropout ratio of 0.5 after word embedding layers and BiLSTM layers. The batch size is set to 16. And the hidden size is set to 128 and 64 for 20News and MRD, respectively. For all other benchmarks, the hidden size is set to 16 to follow the original GCN paper. We use Adam (Kingma & Ba, 2014) as the optimizer. For the text benchmarks, we set the learning rate to 1e-3. For all other benchmarks, we set the learning rate to 0.01 and apply L2 norm regularization with weight decay set to 5e-4. Below we show the hyperparameter associated to IDGL for all benchmarks. All hyperparameters are tuned on the development set.

Table 7: Hyperparameter associated to IDGL on all benchmarks.

| Benchmarks | $\lambda$ | $\eta$ | $\alpha$ | $\beta$ | $\gamma$ | k | $\epsilon$ | m | $\delta$ | T |
|---|---|---|---|---|---|---|---|---|---|---|
| Cora | 0.9 | 0.1 | 0.2 | 0.0 | 0.0 | – | 0 | 4 | 4e-5 | 10 |
| Citeseer | 0.6 | 0.5 | 0.4 | 0.0 | 0.2 | – | 0.3 | 1 | 1e-3 | 10 |
| Wine | 0.8 | 0.7 | 0.1 | 0.1 | 0.3 | 20 | 0.75 | 1 | 1e-3 | 10 |
| Cancer | 0.25 | 0.1 | 0.4 | 0.2 | 0.1 | 40 | 0.9 | 1 | 1e-3 | 10 |
| Digits | 0.4 | 0.1 | 0.4 | 0.1 | 0.0 | 24 | 0.65 | 8 | 1e-4 | 10 |
| 20News | 0.1 | 0.2 | 0.5 | 0.01 | 0.2 | 450 | 0.3 | 10 | 1e-3 | 10 |
| MRD | 0.5 | 0.9 | 0.4 | 0.05 | 0.2 | 350 | 0.3 | 2 | 1e-3 | 10 |

