# OpenReview forum: "Iterative Deep Graph Learning for Graph Neural Networks"
_ICLR.cc/2020/Conference — Reject_

### Official Review · AnonReviewer3 · 2019-10-23
**Official Blind Review #3**

**Rating:** 3

**Review:**

This paper leverages metric learning to learn graph structure jointly with the learning of graph embedding. Firstly, it defines the similarity between any pair of nodes as the cosine similarity of nodal representations learned from attributes from nodes. Some tricks such as multi-head and sparsification are applied to learned cosine similarity to enhance the performance. Secondly, the authors introduce several graph regularizations to make the learned graph smooth, connected, sparse and non-trivial. Finally, the learned graph is linearly combined with the existing graph, using a hyperparameter \lambda. The convergence and time complexity are analyzed. The authors design experiments on five datasets. The paper also contains some issues:

1. Actually, the proposed framework is learning an extra graph adjacency matrix from nodal features, and further train GNN jointly on those two graphs. Therefore, the analysis of \lambda is necessary. However, this part is missing in the paper.

2. The improvements comparing with LDS is not significant. Besides, LDS only uses the optimized graph structure to train GNN, while the proposed framework use both learning structure and the original one (or the kNN result). It raises the question if the proposed framework can still out-perform LDS if LDS also takes the original graph as input to train GNN.

3. The learned graph should be better interpreted. For example, the cosine similarity on citation graphs with sparse features is very likely to be zero. As a result, the learned graph can be extremely sparse with very few non-zero entries. It would be interesting.


**Experience Assessment:**

I have published one or two papers in this area.

**Review Assessment: Checking Correctness Of Derivations And Theory:**

I carefully checked the derivations and theory.

**Review Assessment: Checking Correctness Of Experiments:**

I assessed the sensibility of the experiments.

**Review Assessment: Thoroughness In Paper Reading:**

I read the paper thoroughly.

---

> ### Author Response · Authors · 2019-11-11
> **Author response to Review #3**
>
> We thank the reviewer for providing valuable feedback! Please refer to the response to Reviewer #1 for a detailed statement on the main contributions of this work.
>
> Below we address the concerns mentioned in the review:
>
> 1) The analysis of \lambda is necessary. However, this part is missing in the paper.
>
> A hyperparameter \lambda is used to balance the trade-off between using the learned graph structure and the initial graph structure. Our preliminary experiments showed that it is harmful to totally discard the initial graph structure and using this type of “skip-connection” to incorporate the initial graph structure is more effective than using the initial graph structure as an attention mask, as done in Graph Attention Networks. Below we show the results of using different values of \lambda on Cora. However, in practice, we use validation set to fine tune this hyperparameter and all hyperparameters are listed in our Appendix - Sec. B. Model Setting.
>
> IDGL on Cora:
> Models \ \lambda | 0.9 | 0.8 | 0.7 | 0.6 | 0.5
> IDGL | 83.6 (0.4) | 84.5 (0.3) | 83.9 (0.3) | 82.4 (0.1) | 80.9 (0.2)
>
>
> 2) The improvements comparing with LDS is not significant.
>
> Our model outperforms LDS in 4 out 5 datasets in the transducting setting as shown in Table 1. Our model IDGL is comparable when the initial graph is available (Cora and Citeseer). For the datasets (Wine, Cancer, and Digits) when only initial node features are available, our model consistently achieved better accuracy with much smaller standard deviations. More importantly, one significant advantage of our model compared to LDS is that it can handle inductive setting while LDS cannot as shown in Table 2.
>
> 3) LDS only uses the optimized graph structure to train GNN.
>
> In fact, LDS uses the original graph structure (or the kNN one if the original one is unavailable) to initialize the edge probabilities of the underlying graph. Please see Algorithm 1 Line #3 in the original LDS paper. We have the same settings as LDS in the experiments, and our comparisons are indeed fair.
>
> 4) The learned graph should be better interpreted. For example, the cosine similarity on citation graphs with sparse features is very likely to be zero. As a result, the learned graph can be extremely sparse with very few non-zero entries. It would be interesting.
>
> Yes. Just as the reviewer said, the raw node features of the citation graphs (e.g., Cora and Citeseer) are very sparse. And it is challenging to learn meaningful graph structures solely based on such sparse and indistinguishable features that might not contain enough information about the graph structures. This actually motivates us to propose our iterative graph learning framework where the core idea is to learn better graph structures based on better node embeddings, and in the meanwhile, to learn better node embeddings based on better graph structures.
>
> 5) The authors design experiments on five datasets.
>
> We actually conducted experiments on 7 datasets (5 transductive datasets + 2 inductive datasets) instead of 5 datasets.

---

### Official Review · AnonReviewer1 · 2019-10-23
**Official Blind Review #1**

**Rating:** 6

**Review:**

The paper introduces an iterative method called IDGL for learning both the graph structure (more precisely adjacency matrix) and parameters of the graph neural network.

The idea of iteratively refining an adjacency matrix A to obtain sparsity and smoothness is interesting and the experimental results are quite supportive. The main issue here is that the regularization terms in Eqs. 4, 5, 6 (which should be considered to be the most important in the paper) are exactly similar to those in [1] (see Eq. 12 in [1]).  This reduces the novelty of the paper.

Other parts in the paper such as using similarity between nodes (e.g. self-attention) to compute adjacency matrix or using the learned adjacency matrix with graph neural network is not new and have been done by many other works. There is also the rich related literature on graph generation (e.g., as in drug design), graph transformation (e.g., as in chemical reaction), structure learning in classical probabilistic graphical models, graph pooling (which is essentially building new latent graphs from an original graph), knowledge-graph completion, etc. This is not to say that the problem is solved (it isn't), but it is fair to place this work in a broader context.

About the experiments, I have several concerns. First, I am not sure why the authors say that LDS [2] does not support inductive learning? LDS uses input node features to learn the unknown graph structure so I think it should be able to do inductive learning. DeepWalk or Node2Vec are examples of transductive methods because they do not use the node features. Second, for the running time comparison between IDGL and LDS, what are the size and number of parameters used in each model since they greatly affect the running time.

Minor point: please clean up duplicates in the reference list.

[1] How to learn a graph from smooth signals, Kalofolias et. al. 2016
[2] Learning discrete structures for graph neural networks, Franceschi et. al. 2019.



**Experience Assessment:**

I have published in this field for several years.

**Review Assessment: Checking Correctness Of Derivations And Theory:**

I assessed the sensibility of the derivations and theory.

**Review Assessment: Checking Correctness Of Experiments:**

I assessed the sensibility of the experiments.

**Review Assessment: Thoroughness In Paper Reading:**

I read the paper at least twice and used my best judgement in assessing the paper.

---

> ### Author Response · Authors · 2019-11-11
> **Author response to Review#1**
>
> We thank the reviewer for giving valuable feedback! However, there are some points of misunderstanding that we address in this rebuttal.
>
> We emphasize at the outset that the main contribution of this work is the iterative learning of graph structures and graph node embeddings, which iteratively learn a better graph structure with the updated node embeddings, and learn better node embeddings with the updated graph structure. To the best of our knowledge, we are the first to successfully apply the idea of iterative learning in the literature of graph learning.
>
> In addition, our method dynamically stops when the learned graph structure approaches close enough to the optimal graph based on our proposed stopping criterion. Compared to using a fixed number of iterations globally, the advantage of applying our strategy becomes more clear when we are doing mini-batch training since we can adjust when to stop dynamically for each example graph in the mini-batch.
>
> Compared to LDS that can only handle the transductive learning setting, our model can additionally handle the inductive learning set. We conducted extensive experiments to verify the effectiveness of the iterative learning idea (see Sec 3.4 ablation study) and the dynamic stopping strategy (see Sec 3.5 for model analysis). We also theoretically analyzed and empirically examined the convergence of the proposed iterative learning method. Compared to LDS, our model outperforms it in 4 out of 5 benchmarks.
>
> Below we address the concerns mentioned in the review:
>
> 1) The main issue here is that the regularization terms in Eqs. 4, 5, 6 (which should be considered to be the most important in the paper) are exactly similar to those in [1] (see Eq. 12 in [1]).  This reduces the novelty of the paper.
>
> Graph regularization terms play an important role in our proposed model, however, as we have clarified above, this component should not be considered as the most important for this work. As we mentioned in Sec. 2.2, we borrowed effective techniques in Eqs. 4, 5, 6 from graph signal processing literature (with appropriate citations [1]) and adapted them to our model to regularize the learned graph structure. Compared to [1] that directly learns a graph topology using these techniques, our proposed strategy of training a graph learning model with a joint loss combining this graph regularization loss with the task-specific prediction loss is still novel.
>
> 2) I am not sure why the authors say that LDS [2] does not support inductive learning?
>
> In the original LDS paper, the authors evaluated LDS only in the transductive setting and admitted that “Adding additional nodes after training (the inductive setting) would currently require retraining the entire model from scratch.” From our understanding, the restriction to the transductive setting is because LDS aims at directly optimizing the discrete probability distribution on the edges of the underlying graph (i.e., all pairs of nodes in the graph).  Hence, it cannot handle unseen nodes/graphs in the testing phase. Unlike LDS, our model instead optimizes a shared learnable similarity function between any two node features, which thus can be used to construct a graph for any set of new nodes.
>
> 3) For the running time comparison between IDGL and LDS, what are the size and number of parameters used in each model since they greatly affect the running time.
>
> On the Cora data, the number of trainable parameters of IDGL is 28,836, and for LDS, it is 23,040. So they are comparable in terms of model size. We actually reported the training time of IDGL w/o IL (instead of IDGL) by mistake. Below is the corrected training time (mean and std.) for various models. IDGL is consistently faster than LDS, but in general, they are comparable. We have corrected this part in the revision.
>
>  Models / Benchmarks | Cora | Citeseer | Wine | Cancer | Digits
>   GCN  | 3 (1) | 5 (1) | -- | -- | --
>   GAT   | 26(5) | 28(5) | -- | -- | --
>   LDS | 390 (82) | 585 (181) | 33 (15) | 25 (6) | 72 (35)
>   IDGL | 237 (21) | 563 (100) | 20 (7) | 21 (11) | 65 (12)
>   IDGL w/o IL | 49 (8) | 61 (15) | 3 (2) | 3 (1) | 2 (1)
>
> 4) More related work
>
> We thank the reviewer to referring us to several relevant and interesting works. We could do better by placing our work in a broader context.
>
> 5) Minor point: please clean up duplicates in the reference list.
>
> Yes, we will fix this.

---

### Official Review · AnonReviewer2 · 2019-10-23
**Official Blind Review #2**

**Rating:** 3

**Review:**

The paper proposes an extension of learning graph structure and GNN concurrently, by considering that real-world graphs are often noisy and incomplete. The idea of optimizing the intrinsic graph structure iteratively for down-stream prediction tasks is interesting. Experimental results demonstrate the effectiveness of proposed method.

Strengths:
1）the paper proposes a learnable similarity metric function and a graph regularization for learning an optimal graph structure for prediction.
2）Besides raw node features, the paper attempts to optimize graph structures via learned node embeddings in an iterative manner.
3）The paper is easy to read, and experiments show that the proposed method performs well.

Weaknesses:
1）Compared with LDS [1], this work seems to overlook the bi-level optimization problem for learning model parameters based on the optimal graph structure. The reason behind this method is expected.
2）Although the paper claims that the dependence of raw node features for learning graph structure has been weakened,  empirical analysis on this point is not given. The feature matrices in experiments are not strictly independent with graph structures.
3) As shown in Appendix B, too many hyper-parameters are involved. I conjecture it will be difficult to reproduce the experimental results.
4) Eqs.(2), (3) and (10) are problematic. Node embeddings Z should be included in them. Eq.(10) does not have theoretical proof. According to Eq.(10), the method cannot handle graphs with noisy edges. In experiments, there are edge deletions, but no edge addings. Experiments with attacked graph are expected.
5) Although this method is claimed efficient, it is indeed slower than the classic GNNs due to the iterative operation. The details of training time comparison between this method and GNNs such as GCN and GAT will be helpful. I was wondering why this method is faster than LDS. Is it due to removing the bi-level optimization problem ?
6) Although the method can handle inductive training, it is hardly scale to big networks. Pubmed is an open citation network with around 20,000 nodes similar to Cora and Citeseer. Those three datasets are popularly used in GNNs as testbed. However, Pubmed is not used in this work. I conjecture that the new method cannot handle such a big dataset efficiently.

Overall, this proposed method is well motivated, but the technical novelty is limited.


**Experience Assessment:**

I have published one or two papers in this area.

**Review Assessment: Checking Correctness Of Derivations And Theory:**

I assessed the sensibility of the derivations and theory.

**Review Assessment: Checking Correctness Of Experiments:**

I assessed the sensibility of the experiments.

**Review Assessment: Thoroughness In Paper Reading:**

I read the paper thoroughly.

---

> ### Author Response · Authors · 2019-11-11
> **Author response to Review #2**
>
> We first thank the reviewer for your valuable feedback! Please refer to the overall responses and the responses to Reviewer #1 for a detailed statement on the main contributions of this work.
>
> Below we address the concerns mentioned in the review:
>
> 1) Compared with LDS, this work seems to overlook the bi-level optimization problem
>
> Unlike LDS, in our model, we optimize a joint loss combining both task-specific prediction loss and graph regularization loss. It will be very interesting to see if our model can benefit from adopting the bi-level optimization technique, which we will leave it as future work. However, one severe limitation of the LDS work is that it essentially only optimizes the edge connectivities of the graph assuming the set of nodes are known, which makes it unable to handle the new set of nodes during the testing (the inductive setting).
>
> 2) The feature matrices in experiments are not strictly independent with graph structures
>
> We are sorry about the confusion made here. We are not claiming that the raw node features are independent with graph structures, which is not the focus of this work. Our rationale is as follows. Some previous works rely solely on raw node features to learn the graph structure based on some attention mechanism, which we think have some limitations since raw node features might not contain enough information for learning good graph structures. In this work, we propose a novel iterative learning framework that is able to learn better graph structures with updated node embeddings, and in the meanwhile, learn better node embeddings with updated graph structures. Empirical experiments verify the effectiveness of additionally learning graphs with updated node embeddings. We have updated our manuscript to make it more clear.
>
> 3) As shown in Appendix B, too many hyper-parameters are involved. I conjecture it will be difficult to reproduce the experimental results.
>
> We will release the code and the preprocessed data upon the acceptance of this paper in order to promote the reproducibility of our work. Also, we have listed all hyperparameters associated to IDGL on all benchmarks in Appendix - Sec. B MODEL SETTINGS in our original submission.
>
> 4) Eqs.(2), (3) and (10) are problematic. Node embeddings Z should be included in them. Eq.(10) does not have theoretical proof. According to Eq.(10), the method cannot handle graphs with noisy edges. In experiments, there are edge deletions, but no edge addings. Experiments with attacked graph are expected.
>
> It seems like that there are some misunderstandings here mainly because our notations are confusing. We have made it more clear in our revision. In particular, In Eqs. (1), (2), the two vectors (originally denoted as x_i and x_j and now we renamed them as v_i and v_j to avoid confusion) are served as the inputs for our similarity score function. Note that these two vectors could be any vectors such as raw node features or computed node embeddings. Therefore, Eqs (1), (2), (3) and (10) are all correct.
>
> Eq. (10) is based on our assumption that the optimal graph structure is potentially a shift from the initial graph structure, and our goal is to learn this shift. In particular, it could be interpreted as some form of the skip-connection where a hyperparameter \lambda is used to balance the trade-off between using the learned graph structure and the initial graph structure. Our preliminary experiments showed that it is harmful to totally discard the initial graph structure, and using this type of “skip-connection” to incorporate the initial graph structure with learned graph structure is much more effective.
>
> We do not quite understand the comment made by the reviewer that “according to Eq.(10), the method cannot handle graphs with noisy edges”. We hope the reviewer can kindly clarify this point. Literally, our method is exactly used to enable GNN to cope with graphs with noisy edges or incomplete edges.
>
> However, as suggested by the reviewer, we have done additional experiments on Cora and Citeseer with randomly added edges. Below are the results for test accuracy (std). Through experiments, we found that adding random edges is more challenging than removing random edges. And our model is much more robust than GCN in this scenario. We have added these results in the updated manuscript.
>
> Cora:
> Methods \ added edges percentage | 0% | 25% | 50% | 75%
> GCN | 81.0 (0.2) | 27.6 (8.6) | 20.5 (9.5) | 17.1 (7.7)
> IDGL | 84.5 (0.3) | 62.9 (2.3) | 62.4 (0.9) | 61.7 (1.7)
>
> Citeseer:
> Methods \ added edges percentage | 0% | 25% | 50% | 75%
> GCN | 70.9 (0.3) | 20.6 (3.0) | 20.9 (2.8) | 19.9 (2.7)
> IDGL | 74.1 (0.2) | 60.4 (2.2) | 62.9 (2.5) | 60.6 (1.6)

---

> ### Author Response · Authors · 2019-11-11
> **Author response to Review #2 (continued)**
>
> 5) Although this method is claimed efficient, it is indeed slower than the classic GNNs due to the iterative operation. The details of training time comparison between this method and GNNs such as GCN and GAT will be helpful.
>
> Yes, this is indeed true in terms of run time performance. But note that when input graph structures are noisy or even not available, existing GNNs such as GCN and GAT cannot perform well or even cannot work. We proposed our learning framework IDGL to exactly overcome these limitations.
>
> Nevertheless, based on the reviewer's comments, we conducted additional experiments and reported the training time for all these models. As we explained to the reviewer #1 as well, we previously reported the training time of IDGL w/o IL (instead of IDGL) by mistake. Below is the corrected training time (mean and std.) for various models. IDGL is consistently faster than LDS, but in general, they are comparable. And yes, both IDGL and LDS are slower than GCN and GAT, which is expected since they did not need to learn graph structure simultaneously. We have also updated this part in the revision.
>
>  Models / Benchmarks | Cora | Citeseer | Wine | Cancer | Digits
>   GCN  | 3 (1) | 5 (1) | -- | -- | --
>   GAT   | 26(5) | 28(5) | -- | -- | --
>   LDS   | 390 (82) | 585 (181) | 33 (15) | 25 (6) | 72 (35)
>   IDGL | 237 (21) | 563 (100) | 20 (7) | 21 (11) | 65 (12)
>   IDGL w/o IL | 49 (8) | 61 (15) | 3 (2) | 3 (1) | 2 (1)
>
> 6) Pubmed is not used in this work. I conjecture that the new method cannot handle such a big dataset efficiently.
>
> We admit that scalability might be an issue when applying this method to large networks. It becomes challenging to compute similarity scores for any pair of N nodes when N is a large number. However, this is also a challenge for other graph learning methods. Fo example, in the original LDS paper, the authors admitted that it cannot currently scale to large datasets (due to at least quadratic complexity of node number N). Following the LDS work, we did not evaluate our model on Pubmed either. Note that our current graph learning component (using pairwise similarity metric) can be replaced by other fast and scalable graph learning modules, which we leave it as one of the future works.
>
> 7) I was wondering why this method is faster than LDS.
>
> Please see the above response. The run times are actually comparable, though our approach is slightly faster.

---

### Author Response · Authors · 2019-11-11
**Overall responses for clarifying the main contributions of this work**

We thank all reviewers for their thorough reading and valuable comments. Before we address the specific technical questions from each reviewer, we would like to firstly focus on clarifying the key contributions of this work (our fault for not being more clear).

Very different from the main baseline method - LDS (Franceschi et al., ICML 2019), which jointly learns graph structure and graph node embeddings by learning joint probability distribution on the edges of the graph, we achieve this goal by proposing a novel learning framework consisting of three key components:
    (a) Iterative learning framework to refine the graph structures and graph embeddings.
    (b) Graph learning as similarity metric learning;
    (c) Graph regularization to control smoothness, sparsity, and connectivity;
Among them, the first component (a) is the first time to be proposed (to the best of knowledge) and thus is the most important contribution. The rationale of (a) is to achieve i) refining the adjacency matrix with the updated node embeddings; ii) refining the node embeddings with the updated adjacency matrix. The components of (b) and (c) are combined to learn the graph structure with controlled sparsity and connectivity, which also play an important role in the final performance.

---

### Decision · Program_Chairs · 2019-12-19

**Decision:**

Reject

**Comment:**

The submission proposes a method for learning a graph structure and node embeddings through an iterative process. Smoothness and sparsity are both optimized in this approach. The iterative method has a stopping mechanism based on distance from a ground truth.

The concerns of the reviewers were about scalability and novelty. Since other methods have used the same costs for optimization, as well as other aspects of this approach, there is little contribution other than the iterative process. The improvement over LDS, the most similar approach, is relatively minor.

Although the paper is promising, more work is required to establish the contributions of the method. Recommendation is for rejection.